# Sulforaphane and Its Protective Role in Prostate Cancer: A Mechanistic Approach

**DOI:** 10.3390/ijms24086979

**Published:** 2023-04-10

**Authors:** James Mordecai, Saleem Ullah, Irshad Ahmad

**Affiliations:** 1Department of Bioengineering, King Fahd University of Petroleum and Minerals (KFUPM), Dhahran 31261, Saudi Arabia; 2Department of Transdisciplinary Science and Engineering, School of Environment and Society, Tokyo Institute of Technology, Nagatsuta, Midori-ku, Yokohama 226-8503, Japan; 3Interdisciplinary Research Center for Membranes and Water Security, King Fahd University of Petroleum and Minerals (KFUPM), Dhahran 31261, Saudi Arabia

**Keywords:** sulforaphane, prostate cancer, cancer therapeutics, green chemoprevention, broccoli, cell cycle, myrosinase

## Abstract

The increasing incidence of prostate cancer worldwide has spurred research into novel therapeutics for its treatment and prevention. Sulforaphane, derived from broccoli and other members of the *Brassica* genus, is a phytochemical shown to have anticancer properties. Numerous studies have shown that sulforaphane prevents the development and progression of prostatic tumors. This review evaluates the most recent published reports on prevention of the progression of prostate cancer by sulforaphane in vitro, in vivo and in clinical settings. A detailed description of the proposed mechanisms of action of sulforaphane on prostatic cells is provided. Furthermore, we discuss the challenges, limitations and future prospects of using sulforaphane as a therapeutic agent in treatment of prostate cancer.

## 1. Introduction

The gland known as the prostate is located in the male reproductive system just behind the bladder and surrounds the urethra. The unrestricted proliferation of cells of the prostate gland results in prostate cancer [1]. Prostate carcinoma is one of the most prevalent forms of cancer in men globally, accounting for about 1.4 million new cases and 375,000 mortality per year worldwide [2]. Factors that increase the risk of developing prostate cancer include: age, genetics and lifestyle habits [3]. Considering how common prostate cancer is, the scientific community has intensified efforts in the search for novel therapeutics from naturally occurring compounds capable of preventing, inhibiting or reversing tumor development. Plants have been extensively screened for phytochemicals with anticancer properties; one such phytochemical is sulforaphane [4]. 

Sulforaphane is a small chemical compound found in cruciferous vegetables of the *Brassica* genus (broccoli, broccoli sprouts, kale, cabbage, Brussel sprouts and cauliflower). It is produced when the vegetable is chopped, chewed, boiled or disrupted, causing a plant enzyme myrosinase (EC 3.2.1.147) to convert a precursor molecule called glucoraphanin into sulforaphane. This process also occurs in the human body after consumption of the vegetables, as the gut microbiome contains bacteria that produce myrosinase [5] (Figure 1). 

In the 1990s, sulforaphane was isolated from broccoli for the first time and shown to possess anticancer properties by researchers at Johns Hopkins School of Medicine [6,7]. Subsequently, there has been a plethora of studies reporting the antineoplastic activity of sulforaphane. Recent studies have demonstrated that sulforaphane can prevent the development of cancer cells and initiate apoptosis in a variety of cancer types, including prostate cancer [8]. This is due to the compound’s ability to target multiple signaling pathways involved in cancer cell growth and survival [9,10,11]. Hence, the goal of this review is to present an up-to-date assessment of sulforaphane’s effect on prostate cancer, and to give detailed descriptions of the various proposed mechanisms of action of sulforaphane in the prevention of the progression of prostatic tumors. Challenges and prospective future directions in the use of sulforaphane as a chemo-preventive therapeutic are also discussed. 

## 2. Sulforaphane in Prevention of the Progression of Prostate Cancer

### 2.1. In Vitro Studies

Prostate carcinogenesis is usually initiated by androgen receptor (AR) signaling, and proliferation of cancer cells is promoted by a preferential increase in aerobic glycolysis. A recent study evaluated the shielding properties of sulforaphane and capsaicin against the effect of androgen receptor (AR) stimulators. The researchers manipulated the levels of AR stimulators Androgen and Tip60 by overexpressing these stimulators in LNCaP cells [12]. This resulted in increases in androgen receptors and prostate-specific antigens (PSA), stimulation of AR pathway and proliferation of LNCaP cells by 80–100%. HIF-1α levels were also raised by 52%, which promoted glycolysis. However, 10 μM of sulforaphane totally suppressed the rise and increases brought on by Tip60 and androgen in LNCaP cells. The compound also effectively stopped the increase in both cytosolic and nuclear levels of HIF-1α, reducing glycolysis by 74%. The study concluded that sulforaphane has the ability to reduce Tip60 and androgen-induced proliferation and glycolysis in prostatic tumor cells [12]. 

The impact of sulforaphane and other isothiocyanates on prostate cancer cell lines was assessed in a study that showed that when prostatic carcinoma cell lines PC-3 and DU 145 were treated with 30 μM of sulforaphane for 72 h, it reduced the viability of the cells by 40–60% and inhibited the proliferation of the cell lines [13]. Sulforaphane also decreased the metastatic ability of the cells by up to 50%. When the cell lines were subjected to a combination therapy of sulforaphane and the chemotherapeutic drug Docetaxel (DOCE), the treatment was significantly more efficacious than sulforaphane or DOCE alone. The researchers found that sulforaphane made both cell lines more responsive to DOCE by a synergic mechanism [13]. In a different study, the effect of sulforaphane on PC-3 prostate cancer cells and HDFa normal cells was examined. The study showed that sulforaphane inhibited DNA replication and caused DNA damage in both prostate cancer and normal cell lines. DNA damage, in the form of double-stranded breaks, was more pronounced in cancer cells due to their inability to carry out proper DNA repair. This led to apoptotic elimination of cancer cells [14]. 

Another study reported the anti-tumor properties of sulforaphane on DU145 and PC3 prostatic tumor cell lines via a blockade of cell cycle. A total of 1–20 μM of sulforaphane inhibited the growth of DU145 and PC3 cells. A total of 10 μM of sulforaphane reduced proliferation after 48–72 h of incubation, while 20 μM completely blocked cell growth. Clones were completely destroyed when exposed to 10 μM of sulforaphane for 10 days. Sulforaphane suppressed the multiplication of these prostate cancer cell lines by prompting an arrest of the cell cycle at the S and G2/M phases. This was evident from the increased levels of the proteins responsible for the regulation of cell-cycle, such as CDK1, CDK2 and p19, and from the acetylation of histones H3 and H4 [15]. 

Studies on breast and prostate cancers reported an induction of apoptosis by isothiocyanates [16]. Sulforaphane and two other isothiocyanates induced apoptosis in breast cancer and prostate cancer cell models via ubiquitin proteasome system (UPS)-mediated protein degradation. The study showed that sulforaphane interacts with deubiquitinating enzymes USP14 and UCHL5 and inhibits the activity of these enzymes. When prostate cancer 22Rv1 cells were treated with 25 μM of sulforaphane for 24–36 h, there was an accumulation of poly-ubiquitinated proteins, which prompted protein degradation and eventual apoptosis of the cells. Thus, 80% of viable 22Rv1 cells were lost when exposed to sulforaphane for 24 h [16]. A similar study examined the effect of the induction of autophagy and apoptosis by sulforaphane in PC-3 and castration-resistant 22Rv1 prostate cancer cell lines. The study shows that 10–20 μM of sulforaphane significantly increased lysosome-associated membrane protein 2 (LAMP2) in the cell lines. This induction of LAMP2 levels is undesirable for the prevention of prostatic tumors. However, when LAMP2 was knockdown and the cells were treated with sulforaphane, there was a striking increase in apoptosis in both cell lines. Hence, the study recommended a combination regimen of sulforaphane and a chemical inhibitor of LAMP2 for the chemoprevention of prostate cancer [17] (Table 1).

### 2.2. In Vivo Studies

Prostate cancer is characterized by elevated de novo synthesis of fatty acid and overexpression of key fatty acid synthesis enzymes such as acetyl-CoA carboxylase (ACC) and fatty acid synthase (FASN). Sulforaphane has been shown to prevent prostatic tumor in Transgenic Adenocarcinoma of Mouse Prostate (TRAMP) mice by the inhibition of fatty acid synthesis [18]. Administration of 6 μmol/mouse of sulforaphane to TRAMP mice resulted in 60–70% downregulation of ACC and FASN proteins in prostate tumors and a significant reduction in plasma levels of acetyl-CoA, total free fatty acids and total phospholipids. Human prostate tumors also often exhibit the Warburg phenomenon: a marked increase in aerobic glycolysis. The same group of researchers, in a subsequent study, showed that sulforaphane suppressed glycolysis in prostate neoplastic lesions of mouse models. In the study, two murine models (TRAMP and Hi-Myc) were treated with sulforaphane. When TRAMP mice were given 6 μmol/mouse (1 mg/mouse) three times a week for 17–19 weeks, the prostate tumor expression of glycolysis-promoting enzymes such as hexokinase II (HKII), pyruvate kinase M2 (PKM2) and lactate dehydrogenase A (LDHA) was decreased by 32–45%. Similarly, when Hi-Myc mice were given 1 mg/mouse of sulforaphane three times each week for 5–10 weeks, expression of HKII, PKM2 and LDHA was significantly decreased. These results provide evidence that sulforaphane suppresses in vivo glycolysis in prostate cancer cells [18,19] (Table 1). 

Sulforaphane-rich diets have been shown to reduce the incidence and severity of prostate cancer in TRAMP mice. The study design included TRAMP mice fed a 15% broccoli sprout diet and a control group fed an AIN93G diet for 28 weeks. Tissue samples were collected from these two groups of TRAMP mice at 12 and 28 weeks for examination. At week 28, the group fed with 15% broccoli sprout diet showed a slower rate of prostate tumor development, decreased cancer severity and significant reduction in invasive prostate cancer. Sixteen out of eighteen (89%) control mice had an adenocarcinoma, while just seven out of nineteen (37%) broccoli sprout-fed mice developed adenocarcinoma [20].

**Table 1 ijms-24-06979-t001:** Anti-prostate cancer effects of sulforaphane based on in vitro and in vivo studies.

	In vitro Studies		
Cell Lines	Concentration(Duration)	Anticancer Effects	References
22Rv1	25 μM (24–36 h)	Apoptosis of tumor cells	[17]
LNCaP	10 μM (24–72 h)	Decreased cellular proliferation	[12]
PC-3, DU 145	30 μM (72 h)	Reduced cell viability	[13]
PC-3	40 μM (3–24 h)	DNA damage	[14]
PC-3, DU 145	1–20 μM (48–72 h)	Inhibition of cellular proliferation	[15]
PC-3, 22Rv1	10–20 μM (3–24 h)	Induction of apoptosis and cellular degradation	[16]
	In vivo Studies		
Animal Model	Dose(Duration)	Anticancer Effects	References
TRAMP Mice	15% broccoli sprout diet (12–28 weeks)	Reduced rate of tumor development	[20]
TRAMP Mice	6 μmol, 3-times per week (17–19 weeks)	Downregulation of fatty acid metabolism	[18]
TRAMP Mice, Hi-Myc	6 μmol, 3-times per week (17–19 weeks)1 mg, 3-times per week (5–10 weeks)	Suppression of glycolysis	[19]

### 2.3. Clinical Studies

Investigators carried out a randomized controlled clinical trial (NCT04046653) on 98 men scheduled for prostate biopsy from July 2011 to December 2015. The men were randomly assigned into two groups. One group was given 200 μmol per day of broccoli sprout extract for 4–5 weeks, while the other group received a placebo. At the end of the course of treatment, both groups’ prostate tissues were analyzed for biomarkers and HDAC activity. The study found no significant positive changes in prostate cancer biomarkers. The researchers proposed that the reason for this surprising result could be because the intervention period was short, the dosage was low or insufficient and/or due to the rapid elimination of sulforaphane before it reached the target tissue [21].

A study aiming to evaluate the impact of consuming a glucoraphanin-rich broccoli soup on gene expression in prostate glands of men with localized prostate cancer recruited 49 men diagnosed with organ-confined prostate cancer, who were placed on surveillance to monitor progression of the cancer for a randomized double-blinded controlled trial. The study design randomly divided the 49 participants into a 3-arm intervention. The control arm was given 300 mL portion of broccoli soup made from a standard, commercially available broccoli. The second arm of the study was given the same volume of broccoli soup made from an experimental broccoli genotype enhanced to provide 3 times the glucoraphanin concentration of the control, while the third arm of the intervention received the same volume of broccoli soup that had been enhanced to a glucoraphanin concentration 7 times that of the control. In all arms of the intervention, participants drank 300 mL of these soups weekly for 12 months. Gene expression in the prostate tissues from each patient was quantified by RNA sequencing before and after the dietary intervention. The result of the study indicated an increased level of gene expression consistent with a risk of carcinogenesis in the tissues of participants from the control group. These changes were mildly reduced in the second group, and totally suppressed in the third group. Thus, the study concluded that consuming a glucoraphanin-rich broccoli soup reduces the risk of the progression of prostate cancer [22]. 

A different study tried to explain the mechanism by which sulforaphane affects prostate tissue by showing that sulforaphane and its associated metabolites accumulate in the human prostate gland. Forty-two men scheduled for prostate biopsy were recruited for the study. The study design consists of one placebo and two active interventions: a supplement that provided glucoraphanin (BroccoMax©) and another that provided alliin from garlic. Participants were placed in one of these three groups for 4 weeks. At the end of the intervention period, sulforaphane and alliin levels in biopsy samples from the prostate’s periphery and transition zone were measured. The result of the study shows that the glucoraphanin supplement significantly increased the concentration of sulforaphane and sulforaphane-N-acetyl cysteine in both zones of the prostate gland. It is plausible that this accumulation of sulforaphane in the prostate gland may lead to suppression of prostate cancer progression through a variety of mechanisms [23].

## 3. Mechanisms of Action

### 3.1. AR Signaling

Androgen receptor (AR) signaling mediates the initial stages of prostate carcinogenesis [24]. AR is a hormone receptor and transcription factor. Binding of androgen (such as testosterone) to AR activates AR. The activated AR migrates to the nucleus where it upregulates the transcription of the genes of proteins such as B-cell lymphoma-extra-large (Bcl-XL) and Hypoxia-inducible factor (HIF-1α) [25,26]. Bcl-XL is a protein that suppresses apoptosis and thus promotes the survival and expansion of prostate cancer cells [27]. HIF-1α upregulates the transcription of hexokinase (HK) and pyruvate kinase (PK); over-expression of these enzymes reprograms the metabolism of cells to solely aerobic glycolysis [28]. This is a hallmark of cancer cells known as the Warburg effect [29]. Hence, HIF-1α promotes the development and multiplication of prostate cancer cells via glycolytic metabolism. 

Sulforaphane has been proposed to prevent prostate carcinogenesis by disrupting the AR signaling pathway. Sulforaphane interacts with the promoter region of the AR gene, preventing the transcription of ARs. This significantly reduces the synthesis of ARs, with no ARs present in the cell surface, androgens cannot bind to ARs to initiate the AR signaling cascade [12] (Figure 2). More so, sulforaphane has been shown to suppress HIF-1α [30]. Sulforaphane binds to HIF-1α and distorts its structure; the distorted HIF-1α loses its function and it is subsequently degraded (Figure 2).

### 3.2. Induction of Apoptosis

Apoptosis is a natural occurrence through which the number of cells in tissues are regulated. Cancer develops when apoptosis fails; thus, cancerous tissues often suppress apoptosis in cells [31]. Apoptosis can be induced by the activity of the ubiquitin proteasome system (UPS). The UPS involves two processes: ubiquitination and 26S proteasome-mediated degradation. Improperly folded or damaged proteins are marked by ubiquitin, and then recognized and degraded by 26S proteasome [32]. The 26S proteasome has two subunits: a 20S barrel-shaped catalytic core and a 19S regulatory particle. Deubiquitinating enzymes (DUBs) are attached to the 19S regulatory particle to prevent erroneous degradation of cellular proteins. DUBs remove ubiquitin from poly-ubiquitinated protein, preventing its degradation by the proteasome [33]. Tumor tissues often over-express DUBs such as USP14 and UCHL5, thus preventing degradation of proteins and apoptosis, ultimately resulting in the survival and proliferation of cancerous tissues.

Sulforaphane has been shown to inhibit the two proteasomal cysteine DUBs, USP14 and UCHL5, in prostate cancer cells [16]. Sulforaphane interacts with USP14 and UCHL5 and suppresses their activity. This promotes increased degradation and induces apoptosis of cells of prostate tumor tissue (Figure 3).

### 3.3. DNA Damage

It has been reported that sulforaphane causes double-stranded DNA breaks and then prevents the repair of these breaks in human prostate cancer cells [14,34]. When DNA damage occurs via a double-stranded break, the repair process involves a complex of various nucleotide excision repair proteins. The combined action of MRN and CtIP proteins holds each pair of single DNA strands in place. RPA, BRCA and XPA work together to form a Holliday junction and a primer at the point of repair. Eventually, DNA synthesis is initiated and the damage is repaired [35].

However, in prostate cancer cells, sulforaphane inhibits XPA protein, an important protein involved in nucleotide excision repair. This disrupts and prevents the repair process; multiple double-stranded DNA breaks accumulate in the cell until the cell is destroyed by apoptosis [34] (Figure 4).

### 3.4. Upregulation of Protective Enzymes

Sulforaphane protects against prostate carcinogenesis by upregulating the transcription of carcinogen-detoxifying enzymes (Phase 2 enzymes). Sulforaphane binds to Keap1 in the cytoplasm and disrupts its orientation. This disruption releases Nuclear factor erythroid 2 (Nrf2). Nrf2 is transported to the nucleus, where it binds to antioxidant response element (ARE); this leads to the increased transcription of Phase 2 detoxifying enzymes such as quinone 1, NAD(P)H dehydrogenase and heme oxygenase 1. These enzymes enhance cellular defenses and prevent the initiation of carcinogenesis [5,36] (Figure 5).

### 3.5. Autophagy

Autophagy is a process by which the cell maintains homeostasis by recycling old and damaged cytoplasmic components such as proteins and organelles [37]. The process involves the formation of membranous vacuoles (autophagosomes), which engulf the cytoplasmic components marked for recycling. Autophagosomes fuses with lysosomes, and the lysosomal enzymes degrade the content of the vacuoles [38]. Researchers have shown that autophagy plays a complex role in the development and progression of cancer cells [39].

In the early stages of prostate cancer, sulforaphane induces autophagy by upregulating the transcription of microtubule-associated protein 1 light chain 3 (LC3), an essential protein for the formation of autophagosomes [40]. This induction of autophagy results in cytoprotective effects on prostate cells and the suppression of further progression of prostatic tumors, as damaged and abnormal cell organelles are rapidly degraded [41]. However, in the later stages of the carcinoma, autophagy promotes the survival of cancer cells by shielding them from the effects of stress and therapy. Thus, autophagy inhibitors (such as chloroquine) have been proposed as an adjuvant to sulforaphane for advanced cases of prostate cancer [42]. 

## 4. Limitations and Challenges

As this review has shown, sulforaphane as a natural therapeutic in preventing the progression of human prostate carcinogenesis is very promising and advantageous. However, a number of hurdles and challenges have to be surmounted before it can be used in clinical therapy.

### 4.1. Dosages

There is limited knowledge on the appropriate dosage of sulforaphane that can be administered to humans in a clinical setting. For example, there is a disconnect between doses administered in animal models and allowable doses in humans. Doses ranging from 5 to 100 mg/kg of sulforaphane reduce tumors in animal models [5,19]. For a 70 kg human, this translates to 350–7000 mg/kg, which is significantly above the upper threshold of tolerable doses. As reported by a recent study, administration of low doses of sulforaphane to human subjects shows no positive result [21].

Another limitation is that the therapeutic index of sulforaphane is not known; its range of effective doses and lethal doses has not been worked out. While sulforaphane has been shown to be safe and well-tolerated at low doses, high doses can lead to toxicity and adverse effects. Therefore, it is crucial to standardize the optimal therapeutic dose of sulforaphane. 

In addition, there is an anomaly when the doses of sulforaphane or glucoraphanin used in clinical trials are converted to quantities of raw vegetables to be consumed. The reported average concentration of glucoraphanin in raw broccoli is 0.38 μmol/g [43]. The doses of glucoraphanin used in most clinical trials range from 25 to 800 μmol, which translates to about 65 to 2105 g of raw broccoli. This quantity of raw broccoli cannot be realistically consumed daily.

### 4.2. Bioavailability

There is a dearth of studies on the bioavailability of sulforaphane due to its highly unstable nature. Sulforaphane can be quickly metabolized and eliminated from the body; it becomes difficult to study the bioavailability and pharmacokinetics of the compound. Hence, most studies utilize its precursor glucoraphanin or other forms of its metabolites.

There is also variability in the way individual human subjects metabolize glucoraphanin into sulforaphane; this results in different ranges in the bioavailability of sulforaphane from one subject to the other. Fahey et. al. [44] studied the concentrations of the bioavailable sulforaphane metabolites after administering glucoraphanin to participants. They found a wide range of variability in the ability of individuals to convert glucoraphanin to sulforaphane using their gut myrosinase. The same group of researchers then administered glucoraphanin and myrosinase simultaneously to participants, yet the variability in the conversion and bioavailability of sulforaphane persisted [45]. 

### 4.3. Supplements

As it is practically impossible to match the daily doses of sulforaphane used in clinical trials by eating raw vegetables, supplementation with glucoraphanin or sulforaphane has been recommended. There have been a large number of glucoraphanin/sulforaphane supplements from various companies flooding the market since researchers showed that sulforaphane had anticancer properties and may protect against cancer. Very few of these supplements actually contain sulforaphane and/or glucoraphanin. A few of these supplements have been tested and found not to contain any traces of sulforaphane; some were not extracts of broccoli at all. The few supplements that do contain broccoli extracts faced the challenged of shelve-life and shelve stability, as sulforaphane and myrosinase are highly unstable [46,47]. 

### 4.4. Clinical Trials

The bulk of the studies on the effect of sulforaphane on prostate cancer have been in vitro studies and in vivo studies using animal models. Very few randomized controlled clinical trials have been conducted due to the complexity of conducting one. Some complex factors involved in the design of such a study include: source of sulforaphane (precursors, extracts, supplements or whole vegetables), standardization of efficacious dosage and number and availability of human subjects [46]. Since prostate cancer is a disease of an aged male population, it becomes difficult to recruit a sufficient number of subjects for clinical trials.

As the number of human clinical trials is limited, translation and comparison of the results obtained from animal models to human subjects is not feasible. Extensive knowledge of sulforaphanes’ mechanisms of action and pathways in human subjects in clinical settings is lacking. With such gaps in knowledge, little is known about the long-term effect and off-target effects of chronic use of sulforaphane.

## 5. Conclusions and Future Directions

There is no doubt that sulforaphane has some anticancer and chemoprotective properties. As a natural product it is cheaper and safer than other synthetic anticancer agents. Its potential as a therapeutic agent will continue to spur research.

Future prospects in this area of research should focus on large-scale clinical trials conducted over long periods of time. A standard dosage and the development of a therapeutic index for the use of sulforaphane in clinical settings should also be an area of intense focus.

New systems designed to increase the bioavailability of sulforaphane and improve its absorption by cancer cells are currently being developed. For example, systems such as microencapsulation, microspheres, micelles and nanoparticles will be the direction of future research [48].

Another potential research area may involve human clinical trials designed to assess sulforaphane’s effect on benign prostatic hyperplasia (BPH) and lower urinary tract symptoms (LUTS). As one of sulforaphane’s proposed mechanism of action involves the disruption of AR signaling pathway, and the development of BPH is related to this pathway, sulforaphane can improve symptoms in men with BPH and LUTS. 

In addition, a combination-therapy approach is increasingly being proposed in the treatment of prostate cancer. Combining sulforaphane with other agents, such as chemotherapy and radiation therapy, may enhance its efficacy and should become a staple in future study design. Based on the evidence presented in this review, we conclude that sulforaphane is a promising chemopreventive phytocompound capable of preventing the progression of prostate cancer.

## Figures and Tables

**Figure 1 ijms-24-06979-f001:**
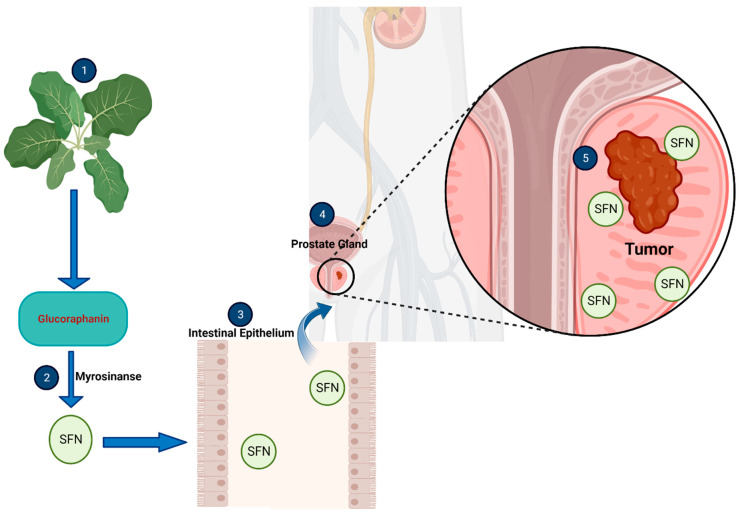
A Synopsis of the Action of Sulforaphane (SFN) on Prostatic Tumor. (**1**) The raw or boiled Cruciferous vegetable is chopped, chewed and eaten. (**2**) Myrosinase converts glucoraphanin to SFN. (**3**) SFN is absorbed and transported into the blood stream. (**4**) SFN accumulates in prostate tumor tissues. (**5**) SFN acts on the tumor. This illustration was made with Biorender.com (accessed on 22 February 2023).

**Figure 2 ijms-24-06979-f002:**
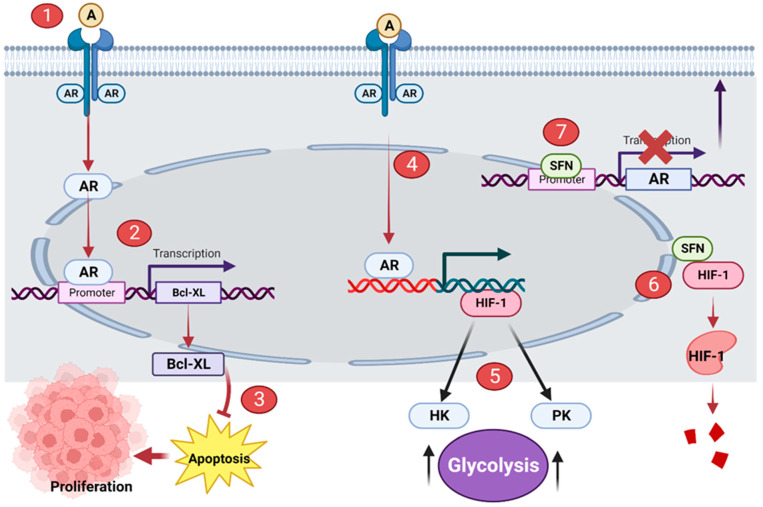
Sulforaphane (SFN) disrupts AR signaling in prostatic tumor cells. (**1**) Androgen (A) binds to AR. (**2**) Activated AR upregulates the transcription of Bcl-XL. (**3**) Bcl-XL suppresses apoptosis and promotes the proliferation of prostate cancer cells. (**4**) Activated AR upregulates the transcription of HIF-1. (**5**) HIF-1 promotes the overexpression of HK and PK; these enzymes initiate the Warburg effect. (**6**) SFN distorts the structure of HIF-1, the distorted protein is subsequently degraded. (**7**) SFN binds to the promoter region of the AR gene, interrupting its transcription and synthesis. This illustration was made with Biorender.com (accessed on 7 March 2023).

**Figure 3 ijms-24-06979-f003:**
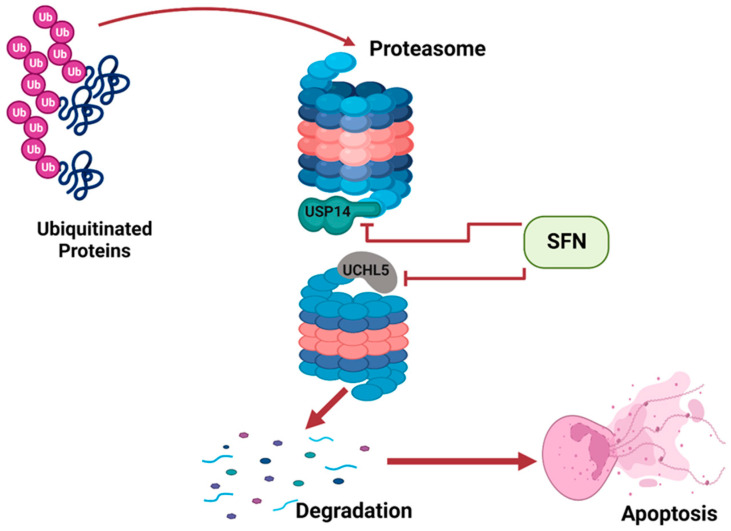
SFN inhibits DUBs of the UPS to induce apoptosis. This illustration was created with Biorender.com (accessed on 7 March 2023).

**Figure 4 ijms-24-06979-f004:**
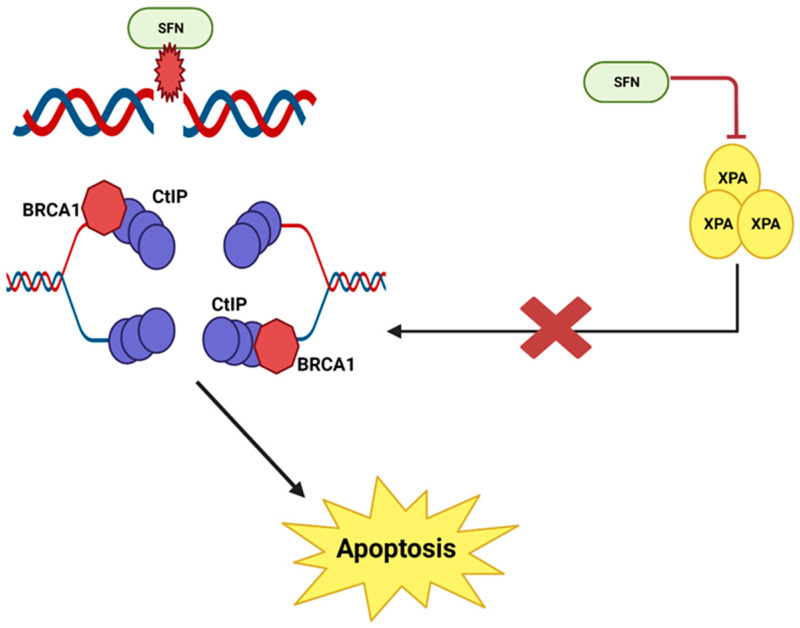
SFN causes DNA damage and prevents DNA repair in prostate cancer cells. This illustration was made with Biorender.com (accessed on 7 March 2023).

**Figure 5 ijms-24-06979-f005:**
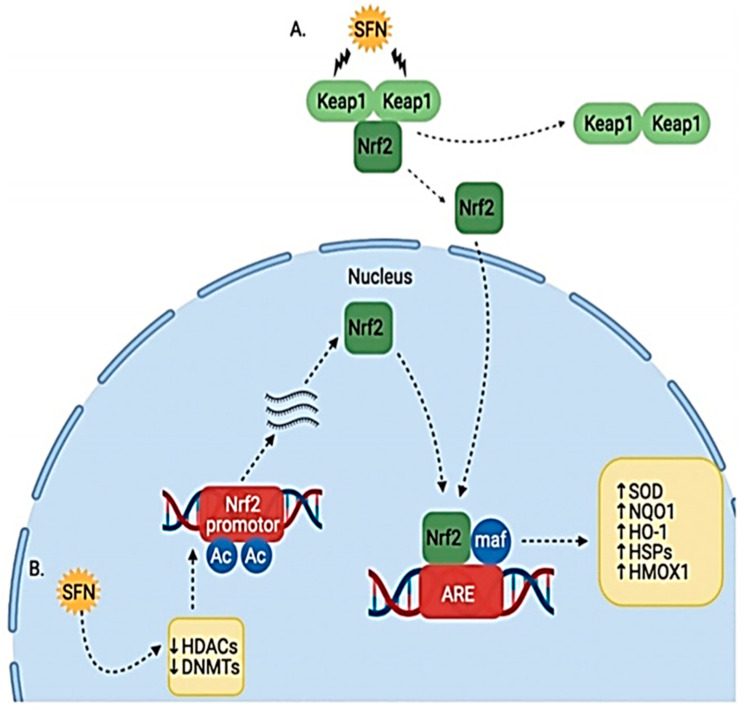
Effect of SFN on Nrf2/Keap1 pathway. (**A**) SFN modifies Keap1 cysteine residues, causing release of Nrf2, which allows it to translocate to the nucleus. (**B**) SFN also induces epigenetic modulation of HDACs and DNMTs, causing increased Nrf2 transcription and translation [5].

## Data Availability

Not applicable.

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
