# Peer review of "Sulforaphane and Its Protective Role in Prostate Cancer: A Mechanistic Approach"

_ijms, 2023, doi:10.3390/ijms24086979_

Round 1

Reviewer 1 Report

This is a well written review article for the effect of Sulforaphane having the prevention effect of prostate cancer. Although, there were many studies including basic and clinical studies for its potential effect for prostate cancer prevention. As a result, the dosage of Sulforaphane is not well established. The figures are well graphed.

I have some questions regarding the manuscript.

1.     There is an empty reference number 42.

2.     Talking about the chemoprevention, the animal studies were used prostate cancer cells. It is for the treatment of cancer instead of prevention. Did any study regarding prevention?

3.     The article described chemoprevention of sulforaphane for prostate cancer regarding blockage of AR pathway. I wonder if it can exert on the benign prostate hyperplasia to decrease the size of prostate in term of treat the LUTS?

4.     Did the sulforaphane reduced level of testosterone in clinical trial?

5.     Did the cell line study include the hormone refractory prostate cancer cell line?

Author Response

Dear Reviewer

Please find attached the document containing point-by-point response of your comments.

Reviewer 2 Report

Dear authors,

Thanks for your contribution to this field. This is an interesting state of art on sulforaphane in Prostate Cancer. The manuscript is still well written and organized.

Nevertheless, all the aspects are almost cover from functions in cells, mouse models or clinical studies to mechanism of action.

Indeed, a part some minor points, my major concern is about the use of “prevention” within the manuscript: title, second paragraph, etc.

Indeed, a molecule which a preventive effect on cancer will avoid disease emergence but, here, sulforaphane is described for its anti-tumor properties.

Please correct within all manuscript.

 Minor points:

Line 125 and 128: can you provide details on this unusual unit mol/mouse or mg/mouse?

Line 148-151: make a proper sentence without bullets.

Lines 261, 280, 293 points before words

 Hope you can address those points.

Best regards.

Author Response

Dear Reviewer

Find attached the document containing point by point response of your comments.

Reviewer 3 Report

This review is very interesting and well constructed. I have few comments for the authors:

- line 27-28, authors should specify if these numbers are worldwide or US

- line 33-34, references are missing here

- line 45-46, references are missing here

- line 64-67, reference is missing here

- line 74-77, reference is missing here

- line 80, reference is missing here

- line 97, reference is missing here

- line 116-117, reference is missing here

- line 142, number of clinical trial is missing here

- Authors should add a section about autophagy because this mechanism might be affected by SFN and studies showed that autophagy play a role in prostate cancer development and resistance to therapies

- Is there any correlation between vegetarian diet and development of prostate cancer compared to people that eat more meat/less vegetables?

Author Response

(The authors gave the same response as above.)
